# Synthesis, Structure, and Electrochemical Properties of 2,3,4,5-Tetraphenyl-1-Monophosphaferrocene Derivatives

**DOI:** 10.3390/molecules28062481

**Published:** 2023-03-08

**Authors:** Almaz A. Zagidullin, Alena R. Lakomkina, Mikhail N. Khrizanforov, Robert R. Fayzullin, Kirill V. Kholin, Tatiana P. Gerasimova, Ruslan P. Shekurov, Ilya A. Bezkishko, Vasili A. Miluykov

**Affiliations:** 1Arbuzov Institute of Organic and Physical Chemistry, FRC Kazan Scientific Center, Russian Academy of Sciences, 8 Arbuzov Street, 420088 Kazan, Russia; 2A.M. Butlerov Chemistry Institute of the Kazan Federal University, 18 Kremlevskaya Street, 420008 Kazan, Russia; 3Department of Physics, Kazan National Research Technological University, 68 Karl Marx Street, 420015 Kazan, Russia

**Keywords:** phosphaferrocenes, electrochemical properties, X-ray structure, DFT calculations, ESR

## Abstract

Heteroleptic 2,3,4,5-tetraphenyl-1-monophosphaferrocene [FeCp(η^5^-PC_4_Ph_4_)] was obtained at a 62% yield through the reaction of lithium 2,3,4,5-tetraphenyl-1-monophosphacyclopentadienide Li(PC_4_Ph_4_) (**1**) with [FeCp(η^6^-C_6_H_5_CH_3_)][PF_6_]. The structure of 1-monophosphaferrocene **2** and its W(CO)_5_-complex **3** were confirmed by multinuclear NMR and single-crystal X-ray diffraction study and further supported by DFT calculations. Cyclic voltammetry demonstrated that [FeCp(η^5^-PC_4_Ph_4_)] **2** has a quasi-reversible oxidation wave. The comparison of the properties of phosphaferrocene **2** with those of W(CO)_5_-complex **3** shows the possibility of changing the coordination type during oxidation.

## 1. Introduction

The discovery of ferrocene [Fe(η^5^-C_5_H_5_)_2_] approximately seventy years ago significantly influenced chemical research and provided a key boost for establishing and expanding organometallic chemistry, which has continued to develop rapidly. Over the years of intensive research, the ferrocene unit has been recognized as an extremely versatile platform for ligand design, materials research, medicinal chemistry, and many other research fields [1]. Among the various heterometallocenes reported to date, monophosphaferrocenes are by far the most investigated [2,3,4]. Recently, a facile one-step method for the synthesis of “fully inorganic” ferrocene analogue was reported and [Fe(P_4_)_2_]^2−^ represents the closest all-phosphorus derivatives of iron to ferrocene [Fe(η^5^-C_5_H_5_)_2_] so far [5]. Phosphaferrocenes are commonly regarded as phosphorus ligands with weaker σ-donor properties than classical tertiary phosphines and stronger π-acceptor ability similar to that of phosphites P(OR)_3_ [6]. From a practical standpoint, monophosphaferrocenes have been utilized as chiral ligands in homogeneous and asymmetric catalysis [7,8,9,10,11,12,13,14], as building blocks for multidentate ligand systems [15,16,17,18], and as functional materials for self-assembled monolayers [19,20].

At present, two main protocols have been developed for the preparation of monophosphaferrocenes. The first is the reaction between P-phenyl-phosphole and [CpFe(CO)_2_]_2_ at high temperatures, which was developed by the Mathey workgroup in 1977 [21,22]. The phosphaferrocenes obtained through this route have a tendency to be contaminated by the 2-phenylated derivative appearing through the thermal [1,5]-sigmatropic shift of the P-phenyl substituent onto the phosphole ring. Therefore, this procedure provides a desired product with low yields [23]. The second method of the synthesis of monophosphaferrocenes is the reaction between monophospholide anion and cationic (π-arene)iron(II) complex. In 1986, Wells demonstrated that [(η^6^-mesitylene)FeCp]PF_6_ complex playing the role of CpFe^+^ synthon is an excellent precursor for the synthesis of monophosphaferrocenes [24]. Generally, phospholide anions are prepared by the reductive cleavage of the exocyclic C–P bond in P-phenyl-1-monophospholes with lithium metal. However, phenyllithium PhLi is an undesirable by-product, the deactivation of which is necessary. This method has recently been modified by the use of inexpensive aluminum chloride as an in situ-generated phenyllithium scavenger, and thus a 50% yield of desired [(η^6^-mesitylene)FeCp]PF_6_ was attained [25].

From the atom-economical point of view, it is better to use ready-made monophospholide anion uncontaminated with phenyllithium, since nucleophilic PhLi reacts with phosphaferrocenes [26,27]. Existing synthetic methods allow various phospholide anions to be obtained in their pure form, without PhLi impurities [4,28]. We have recently reported a convenient and effective method for the preparation of heteroleptic 1,2-diphosphaferrocenes [29,30] and 1,2,3-triphosphaferrocenes [31,32] through the reaction of appropriate 1,2-diphospholide- or 1,2,3-triphospholide anions in their pure form with a [(η^6^-toluene)FeCp]PF_6_ complex. In the present article, we describe the rational and atom-economical synthesis of 2,3,4,5-tetraphenyl-1-monophosphaferrocene and its W(CO)_5_ complex and compare their structural and electrochemical properties with the previously known analogues.

## 2. Results and Discussion

### 2.1. Synthesis and Structure of 2,3,4,5-Tetraphenyl-1-Monophosphaferrocene Derivatives

The target 1-monophosphaferrocene was prepared via a classical two-step sequence. At the first step, the highly moisture-sensitive lithium 2,3,4,5-tetraphenyl-1-monophosphacyclopentadienide (**1**) was obtained by straightforward synthesis from elemental phosphorus P_4_ and in situ-generated 1,4-dilithio-1,2,3,4-tetraphenylbutadiene. Compound **1** was characterized by the ^31^P NMR resonance at +99 ppm. This direct procedure based on elemental (white) phosphorus activation has advantages such as step-economy (two steps in one flask), mild conditions (+25 °C, 2 days), and good yields (up to 63%) [33,34].

In the next step, the lithium phospholide **1** was converted into phosphaferrocene **2** upon reaction with (toluene)cyclopentadienedienyl-iron(II) hexafluorophosphate salt [FeCp(η^6^-C_6_H_5_CH_3_)][PF_6_] at a 1:1 ratio in boiling diglyme in 2 h (Figure 1). Diglyme was evaporated and the product was extracted with toluene. The subsequent filtration of toluene solution through a silica gel layer gave pure 2,3,4,5-tetraphenyl-1-monophosphaferrocene (**2**) as an air-stable powder in satisfactory yields (68–72%). Novel monophosphaferrocene **2** was characterized by multinuclear NMR spectroscopy and elemental analysis (Appendix A). The ^31^P{^1^H} NMR spectrum of **2** shows the singlet at –61 ppm shifted upfield in comparison to 1-monophospholide lithium **1** by ca. 160 ppm. In the ^1^H NMR spectrum, the characteristic signals of the aryl substituents at 7.06–7.20 ppm and the cyclopentadienyl ring at 4.43 ppm can be observed. The ^13^C{^1^H} NMR spectrum of **2** shows doublets at 99 ppm (^1^*J*_PC_ = 57.7 Hz) and 100 ppm (^2^*J*_PC_ = 4.5 Hz) for the carbon atoms of the 1-monophosphacyclopentadienyl ligand and a singlet at 76 ppm for the cyclopentadienyl ligand. The NMR data for 2,3,4,5-tetraphenyl-1-monophosphaferrocene **2** are comparable to those of related compounds bearing alkyl [35,36] or aryl [37,38] substituents.

The structure of **2** was undoubtedly confirmed by the single-crystal X-ray diffraction. Appropriate single crystals were obtained by crystallization from a toluene solution. Complex **2** crystallizes in the orthorhombic space group *Pbca* with a single molecule in the asymmetric cell (Figure 1). The phospholyl (PC_4_) and cyclopentadienyl (C_5_) ligands of **2** are almost eclipsed with a turning angle P1–*Cnt*(PC_4_)–*Cnt*(C_5_)–C5 of 12.38(6)° (*Cnt* is centroid), and their two planes form an angle ∠(PC_4_)(C_5_) equal to 3.14(4)° (Table 1). Selected internuclear distances characterizing the coordination sphere are listed in the caption. The phenyl substituents exhibit a propeller-like arrangement with torsion angles varying from 120.9° to 140.5°. All geometrical parameters (bond angles and bond lengths) of **2** are similar to those of the related monophosphaferrocenes with alkyl substituents (Table 1). It is worth noting that in this series, compound **2** has the shortest Fe–*Cnt*(PC_4_) distance, while Fe–*Cnt*(C_5_) distances are quite close. Despite the steric volume of four phenyl substituents, the smallest Fe–P distance is also observed for **2**.

Tungsten complex **3**, 2,3,4,5-tetraphenyl-1-monophosphaferrocene-1-tungstenpentacarbonyl, was obtained through the reaction of 2,3,4,5-tetraphenyl-1-monophosphaferrocene (**2**) with labile complex W(CO)_5_(THF) at 25 °C with a yield of 86% (Figure 1). It is worth noting that the reaction of **2** with stable complex W(CO)_5_(CH_3_CN) did not proceed at temperatures from 25 to 110 °C. Phosphaferrocene **2** behaves as a weak σ-donor ligand.

In the ^31^P{^1^H} NMR spectrum of **3**, a singlet at –30 ppm with coupling constant ^1^*J*_WP_ = 262 Hz was observed. Both ^1^H and ^13^C NMR spectra confirm a definite structure and purity of complex **3** (Appendix A). In the IR spectrum of **3** recorded in a KBr pellet, four absorption bands ν(CO) were observed at 1930, 1948, 1966, and 2074 cm^−1^, which are characteristic for the W(CO)_5_L complexes. The IR-spectroscopic investigation of **3**, in comparison with [(PPh_3_)W(CO)_5_] and [(2,4,6-triphenylphosphinine)W(CO)_5_], reveals the expected trend of the donor–acceptor capabilities of the corresponding ligands. The CO stretching frequencies ν(CO) in the IR spectra clearly indicated that the 2,3,4,5-tetraphenyl-1-monophosphaferrocene (**2**) (highest ν(CO) = 2074 cm^−1^) and 2,4,6-triphenylphosphinine (highest ν(CO) = 2073 cm^−1^) is the poorest electron pair donors while tripheylphosphine showed CO stretching frequencies at ν(CO) = 2071 cm^−1^ [40]. As expected, the IR studies of these complexes showed that **2** is a better π-acceptor than the 2,4,6-triphenylphosphinine and PPh_3_. These results display a high π-acceptor with poor σ-donor ability of **2**.

### 2.2. Electrochemical Properties of 2,3,4,5-Tetraphenyl-1-Monophosphaferrocene Derivatives

The electrochemical properties of monophosphaferrocenes, especially those containing aryl substituents, remain poorly investigated. According to the literature data, the introduction of one phosphorus atom instead of the CH-fragment in ferrocene leads to higher oxidation potentials compared to ferrocene [Fe(η^5^-C_5_H_5_)_2_] [41,42]. At the same time, the presence of two or more methyl groups has a slight effect on the HOMO–LUMO gap of monophosphaferrocenes (Table 2).

In this work, compounds **2** and **3** were studied by cyclic voltammetry. Compound **2** has a quasi-reversible oxidation wave at a potential of 0.55 V vs. FcH/FcH^+^, which is 0.49 V more anodic than the literary analogue [(Me_5_Cp)Fe(η^5^-PC_4_Ph_4_)] (Figure 2). Despite the paucity of literature data on phosphaferrocenes, it is generally accepted that the presence of one phosphorus atom in the structure of the cyclopentadienide ring does not lead to irreversible oxidation processes in a phosphaferrocene solution.

The quasi-reversibility during the oxidation of structure **2** can be associated with the fact that during the formation of Fe^II^ in Fe^III^, the P-atom could be coordinated to the Fe-atom, as a result of which the re-reduction potential (−0.28 V vs. FcH/FcH^+^) is shifted to the negative region (Figure 2). Previously, the formation of such complexes was demonstrated in the case of phosphanickelocene [43]. A change in the type of coordination can also lead to intramolecular disproportionation, where the charge may not necessarily be stored on the Fe-atom or phospholide ring (Figure 2). This assumption is visually confirmed by comparing the electrochemical properties of compound **2** and its complex **3** with tungsten, in this case of which quasi-reversibility disappears in cyclic voltammetry. Since the lone pair of P-atom is bounded to W-atom, the intramolecular rearrangement of the phospholide becomes impossible, and thus the stabilization of the oxidized Fe-atom becomes unlikely. Additionally, bulky phenyl fragments do not allow the electrolyte anion to move close enough to stabilize the positive charge, as a result of which an irreversible oxidation wave is observed.

It is well known that the oxidation of the ferrocene [Fe(η^5^-C_5_H_5_)_2_] molecule leads to the appearance of a Fe^III^ cation with 3d^5^ configuration in a low spin state [44]. Although low spin state Fe^III^ complexes are often observed by ESR (electron paramagnetic resonance) [45,46] and have a g-factor close to the g-factor of the free electron of 2.0023, the ferrocenium cation is ESR-silent at temperatures above 78 K, which is due to the short relaxation time. Indeed, the oxidation of [Fe(η^5^-C_5_H_5_)_2_] in the electrochemical ESR cell did not lead to the appearance of any signals. At the same time, the oxidation of phosphaferrocene **2** leads to the appearance of a single line with magnetic resonance parameters g = 2.0019 and ΔH = 7 G at a potential of 0.55 V (vs. FcH/FcH^+^) (Figure 3). We attribute this signal to the phosphaferrocenium cation of **2** in the low-spin state since complexes with high-spin Fe^III^ have a much larger line width [47,48]. The oxidation of the W(CO)_5_ complex **3** does not lead to the appearance of an ESR signal, which does not provide an unambiguous answer to the question about the state of Fe^III^ in the oxidized form of **3**. Such behavior of complex **3** can be explained by the assumption that the relaxation time of the cation of **3** is shorter than that of the cation of **2**.

The preference for the low-spin state of oxidized species **2** and **3** was also shown quantum-chemically. Thus, geometries of monophosphaferrocene **2** and its tungsten complex **3** have been optimized quantum-chemically together with their cations (Appendix A). For cations, two possible spin states have been considered, namely *S* = 1/2 (low-spin) and *S* = 5/2 (high-spin). For both low-spin cations, computations predict the elongation of distances between cyclopentadienyl (C_5_) and phospholyl (PC_4_) rings and the Fe-atom. The optimization of the high-spin states of **2** and **3** leads to the notable distortion of structures (Table 3). The substituted phospholyl rings (PC_4_) “tilt” from the initial position. Energetically, for both cationic forms, the low-spin state is more stable compared to the high-spin state.

The presence of four phenyl rings also significantly lowers the reduction potential of the phospholide ring, and as a result, the HOMO–LUMO gap decreases, which makes them thermodynamically more stable. The tungsten complex **3** has two reduction waves, unlike the phosphaferrocene **2** (Figure 4). In the literature [49], the reduction of the W(CO)_5_ complex of 3,3′,4,4′-tetramethyl-1,1′-diphosphaferrocene was accompanied by an electrochemical–chemical mechanism. In our case, with only one phospholide ligand, this mechanism is not implemented, although two reduction waves are also observed, because, in this case, the second reduction wave does not coincide with phosphaferrocene **2**. The first reduction wave can be attributed to the formation of a radical anion on the phospholide anion (Figure 3). The shift of the potential in comparison with **2** to the anodic region is associated with the shift in the electron density from the phosphaferrocene fragment to the W(CO)_5_ fragment. The second reduction wave probably refers to the reduction of the W(CO)_5_ fragment and, under experimental conditions, has time to be fixed without decomposition.

## 3. Materials and Methods

### 3.1. General

The NMR spectra were recorded on a Bruker MSL-400 (^1^H 400 MHz, ^31^P 161.7 MHz, ^13^C 100.6 MHz). SiMe_4_ was used as an internal reference for ^1^H and ^13^C NMR chemical shifts, and 85% H_3_PO_4_ as an external reference for ^31^P NMR. All experiments were carried out using standard Bruker pulse programs. The infrared (IR) spectra were recorded on a Bruker Vector-22 spectrometer.

### 3.2. DFT Calculations

All calculations were performed with the Gaussian 16 suite of programs [50]. The hybrid PBE0 functional [51] and the Ahlrichs’ triple-ζ def-TZVP AO basis set [52] were used for the optimization of all structures. In all geometry optimizations, the D3 approach [53] was applied to describe the London dispersion interactions, as implemented in the Gaussian 16 program.

### 3.3. Electrochemical Measurements

Electrochemical measurements were conducted with a BASi Epsilon EClipse electrochemical analyzer. The program concerned Epsilon-EC-USB-V200 waves. A conventional three-electrode system was used with glassy carbon (GC) or carbon paste electrode (CPE) solutions for powder samples as the working electrode, the Ag/AgCl (0.01 M) electrode as the reference electrode, and a Pt wire as the counter electrode. A 0.1 M Et_4_NBF_4_ was used as the supporting electrolyte to determine the current–voltage characteristics.

### 3.4. ESR Measurements

ESR measurements were carried out on an X-band ELEXSYS E500 ESR spectrometer. Samples in a cell of combined electrochemistry–ESR were inserted into an ER 4102ST cavity, after which the spectrometer was tuned and the ESR spectra were recorded. Oxygen was removed from liquid samples through three cycles of “freezing in liquid nitrogen–evacuation–thawing” and, after the last cycle, the cell was filled with gaseous helium. The material of the auxiliary electrode was platinum, the reference electrode was Ag/AgCl, and a platinum plate served as a working electrode. A Bruker E 035M teslameter was used to accurately determine the g-factor.

### 3.5. Single Crystal X-ray Diffraction

The X-ray diffraction data for the single crystal **2** were collected on a Bruker D8 QUEST diffractometer with a PHOTON III area detector and an IμS DIAMOND microfocus X-ray tube, using Mo *K*α (0.71073 Å) radiation. The diffractometer was equipped with an Oxford Cryostream LT device for low-temperature experiments. The data reduction package *APEX*4 v2021.10-0 was used for data collecting and processing. The analysis of the integrated data did not show any decay. The data were corrected for systematic errors and absorption: numerical absorption correction based on integration over a multifaceted crystal model and empirical absorption correction based on spherical harmonics according to the *mmm* point group symmetry using equivalent reflections. The structures were solved by the direct methods using *SHELXT*-2018/2 [54] and refined by the full-matrix least-squares on *F*^2^ using *SHELXL*-2018/3 [55]. Non-hydrogen atoms were refined anisotropically. The hydrogen atoms were inserted at the calculated positions and refined as riding atoms.

Crystallographic data for **2**. C_33_H_25_FeP, orange prism (0.434 × 0.380 × 0.367 mm^3^), formula weight 508.35 g mol^−1^; orthorhombic, *Pbca* (No. 61), *a* = 12.8590(3) Å, *b* = 14.8048(3) Å, *c* = 25.9145(5) Å, *V* = 4933.47(18) Å^3^, *Z* = 8, *Z*′ = 1, T = 100(2) K, *d*_calc_ = 1.369 g cm^−3^, μ(Mo *K*α) = 0.696 mm^−1^, *F*(000) = 2112; *T*_max/min_ = 0.6842/0.6168; 227694 reflections were collected (2.231° ≤ θ ≤ 32.060°, index ranges: −19 ≤ *h* ≤ 19, −21 ≤ *k* ≤ 22 and −38 ≤ *l* ≤ 38), 8559 of which were unique, *R_int_* = 0.0485, *R*_σ_ = 0.0194; completeness to θ of 32.060° 99.3%. The refinement of 316 parameters with no restraints converged to *R*1 = 0.0310 and *wR*2 = 0.0762 for 7234 reflections with *I* > 2σ(*I*) and *R*1 = 0.0424 and *wR*2 = 0.0810 for all data with goodness-of-fit *S* = 1.039 and residual electron density ρ_max/min_ = 0.410 and –0.544 e Å^−3^, rms 0.064; max shift/e.s.d. in the last cycle 0.004. Deposition number 2218908 contains the supplementary crystallographic data for compound **2**. These data are provided free of charge by the joint Cambridge Crystallographic Data Centre and Fachinformationszentrum Karlsruhe Access Structures service www.ccdc.cam.ac.uk/structures (deposited on 10 November 2022).

### 3.6. Synthesis

All reactions and manipulations were carried out under dry pure N_2_ in the standard Schlenk apparatus. All solvents were distilled from sodium/benzophenone or phosphorus pentoxide and stored under nitrogen before use. Starting materials [FeCp(η^6^-C_6_H_5_CH_3_)][PF_6_] [56] and Li(THF)_2_-2,3,4,5-tetraphenyl-1-monophosphacyclopentadienide (**1**) [34] were prepared according to literature procedures. W(CO)_6_ was purchased from Aldrich and used without additional purification.

#### 3.6.1. Synthesis of 2,3,4,5-Tetraphenyl-1-Monophosphaferrocene (**2**)

[FeCp(η^6^-C_6_H_5_CH_3_)][PF_6_] (0.54 g, 1.51 mmol) was added to lithium 2,3,4,5-tetraphenyl-1-monophospholide (**1**) (0.82 g, 1.52 mmol) in 20 mL of diglyme. The reaction mixture was stirred at 25 °C for 1 h and then heated to 160 °C for additional 2 h. Then, the reaction mixture was cooled to 25 °C, filtered, and the solvent was evaporated and the remaining solid was dissolved in 30 mL toluene. The toluene solution was kept at −20 °C for 2 days, filtered, and passed through a layer of silica (4–5 cm), and the silica was additionally washed with toluene (3 × 15 mL). After the removal of the solvent, compound **2** was obtained as a reddish powder (0.62 g, 72% yield), and recrystallization from hot toluene gave crystals 2,3,4,5-tetraphenyl-1-monophosphaferrocene (**2**) with m.p. 180 °C. ^1^H NMR (CDCl_3_, *δ*, ppm, *J*, Hz): 4.43 (s, 5H, Cp), 7.06–7.20 (m, 20H, Ph). ^31^P{^1^H} NMR (CDCl_3_, *δ*, ppm, *J*, Hz): −60.8 (s). ^13^C{^1^H} (CDCl_3_, *δ*, ppm, *J*, Hz): 75.5 (s, Cp), 99.2 (d, ^1^*J*_PC_ = 57.7, C2/C5), 99.8 (d, ^2^*J*_PC_ = 4.5, C3/C4), 126.0 (s, Ph), 126.5 (s, Ph), 127.2 (s, Ph), 127.4 (s, Ph), 130.5 (d, ^3^*J*_PC_ = 7.1, Ph), 132.4 (s, Ph), 137.0 (s, Ph), 139.2 (d, ^3^*J*_PC_ = 16.8, Ph). IR (KBr, cm^−1^): 460 (w), 493 (w), 562 (w), 591 (w), 697 (s), 718 (s), 747 (m), 759 (w), 825 (w), 916 (w), 1026 (w), 1074 (w), 1156 (w), 1388 (w), 1492 (s), 1597 (w), 1871 (w), 1945 (w), 2345 (w), 2926 (w), 2963 (w), 3054 (w), 3077 (w). Calculated for C_37_H_37_FeP (M 568): C 78.17, H 6.56, Fe 9.82, P 5.45. Found: C 78.33, H 6.72, P 5.71. 

#### 3.6.2. Synthesis of 2,3,4,5-Tetraphenyl-1-Monophosphaferrocene-1-Tungstenpentacarbonyl (**3**)

A solution of W(CO)_6_ (0.35 g, 1.0 mmol) in THF (100 mL) was exposed to UV light (365 nm) in a quartz reaction vessel under argon at 0 °C for 3 h. The color of the resulting solution was yellow. A solution of **2** (0.56 g, 1.0 mmol) in THF was added and the reaction mixture was stirred for 20 h at 25 °C. The color changed to brown-red. The solvent was removed in vacuo and the product was extracted with toluene. The solvent was evaporated to give 0.76 g (86%) **3** as an orange powder with m.p. 204 °C. ^1^H NMR (CDCl_3_, *δ*, ppm, *J*, Hz): 4.70 (s, 5H, Cp), 6.94–7.30 (m, 20H, Ph). ^31^P{^1^H} NMR (CDCl_3_, *δ*, ppm, *J*, Hz): −30.1 (s, ^1^*J*_PW_ = 263.3). ^13^C{^1^H} (CDCl_3_, *δ*, ppm, *J*, Hz): 77.3 (s, Cp), 93.4 (s, C2/C5), 97.2 (s, C3/C4), 126.9 (s, Ph), 127.1 (s, Ph), 127.6 (s, Ph), 127.7 (s, Ph), 132.6 (s, Ph), 132.6 (s, Ph), 132.7 (s, Ph), 136.2 (s, Ph), 136.7 (s, Ph). IR (KBr, cm^−1^): 491 (w), 513 (w), 574 (w), 593 (w), 663 (w), 669 (w), 800 (s), 865 (m), 1020 (br.s.), 1098 (br.s.), 1262 (s), 1414 (w), 1445 (w), 1470 (w), 1496 (w), 1930 (m), 1948 (m), 1966 (w), 2074 (w). Calculated for C_42_H_37_FePO_5_W (M 892): C 56.53, H 4.18, Fe 6.26, P 3.47, W 20.60. Found: C 56.49, H 4.32, P 3.68.

## 4. Conclusions

In this paper, we described the rational synthetic method of novel 2,3,4,5-tetraphenyl-1-monophosphaferrocene **2** and its W(CO)_5_-complex **3** and elucidated their electrochemical properties. The structures were extensively studied from experimental (NMR and IR spectroscopies and X-ray diffraction) and theoretical points of view. Chemical properties and IR study showed a high π-acceptor with poor σ-donor ability of 2,3,4,5-tetraphenyl-1-monophosphaferrocene (**2**). Cyclic voltammetry showed that [CpFe(η^5^-PC_4_Ph_4_)] **2** has a quasi-reversible oxidation wave and a potential more positive by 0.49 V than its literary analogue [(Me_5_Cp)Fe(η^5^-PC_4_Ph_4_)]. A comparison of electrochemical properties with the tungsten complex **3** showed the possibility of changing the type of coordination upon oxidation.

## Data Availability

The data presented in this study are contained within the article or are available upon request from the first author, A.A.Z.

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
