# Peer review of "Synthesis, Structure, and Electrochemical Properties of 2,3,4,5-Tetraphenyl-1-Monophosphaferrocene Derivatives"

_molecules, 2023, doi:10.3390/molecules28062481_

Round 1
Reviewer 1 Report
The authors have synthesized and characterized the novel compound 2, 3, 4, 5-tetraphenyl-1-monophosphaferrocene derivatives. The result can be supported by the X-ray diffraction data. The electrochemical properties are investigated by the cyclic voltammetry. They also proposed oxidation mechanism based on the rational analysis. In addition, the DFT calculations are performed to help further analysis of the results.
The conclusion is correct, and the method is appropriate. The manuscript is generally well written, but minor revision is necessary. Therefore, I recommend publication after minor revision based on the following comments.
1. Recently, the inorganic ferrocene analogue [Fe(P4)2]2– was reported (J. Am. Chem. Soc. 2022, 144, 15, 6698–6702). Some statements can be added to the introduction,.
2.“2,3,4,5-tetraphenyl-1-monophosphaferrocene-1-tungstenpentacarbonyl, was obtained by reaction of 2,3,4,5-tetraphenyl-1-monophosphaferrocene (2) with W(CO)5THF at 25 °C with a yield of 86 % (Scheme 1). It is worth noting that the reaction of 2 with W(CO)5(NCCH3) did not proceed at temperatures from 25 to 110 °C.”
Please further explain the effect of temperature on the reaction and yield
3.The DFT results should be listed the Supporting information. This makes it easy for the reader to repeat.
Author Response
Response to Reviewer 1 Comments
Manuscript ID: molecules-2222464
Title: Synthesis, structure, and electrochemical properties of 2,3,4,5-tetraphenyl-1-monophosphaferrocene derivatives
Reviewer #1 The authors have synthesized and characterized the novel compound 2,3,4,5-tetraphenyl-1-monophosphaferrocene derivatives. The result can be supported by the X-ray diffraction data. The electrochemical properties are investigated by the cyclic voltammetry. They also proposed oxidation mechanism based on the rational analysis. In addition, the DFT calculations are performed to help further analysis of the results. The conclusion is correct, and the method is appropriate. The manuscript is generally well written, but minor revision is necessary. Therefore, I recommend publication after minor revision based on the following comments.
Reply: First of all, we would like to thank the Reviewer for the detailed consideration of our manuscript, questions and remarks.
Q1. Recently, the inorganic ferrocene analogue [Fe(P4)2]2– was reported (J. Am. Chem. Soc. 2022, 144, 15, 6698–6702). Some statements can be added to the introduction
Reply: The authors acknowledge the referee's meaningful remark. The following sentence and reference were added. «Recently, a facile one-step method for the synthesis of «fully inorganic» ferrocene analogue was reported, and [Fe(P4)2]2− represents the closest all-phosphorus derivatives of iron to ferrocene [Fe(η5-C5H5)2] so far [Wang, Z.C.; Qiao, L.; Sun, Z.M.; Scheer, M. Inorganic Ferrocene Analogue [Fe(P4)2]2–. J. Am. Chem. Soc. 2022, 144, 6698-6702. doi:10.1021/jacs.2c01750]»
Q2. “2,3,4,5-tetraphenyl-1-monophosphaferrocene-1-tungstenpentacarbonyl, was obtained by reaction of 2,3,4,5-tetraphenyl-1-monophosphaferrocene (2) with W(CO)5THF at 25 °C with a yield of 86 % (Scheme 1). It is worth noting that the reaction of 2 with W(CO)5(NCCH3) did not proceed at temperatures from 25 to 110 °C.” Please further explain the effect of temperature on the reaction and yield
Reply: The reaction of 2,3,4,5-tetraphenyl-1-monophosphaferrocene 2 with stable complex W(CO)5(NCCH3) did not proceed at temperatures from 25 to 110 °C at all. At the same time the reaction of 2,3,4,5-tetraphenyl-1-monophosphaferrocene 2 with labile complex W(CO)5THF proceed at 25 °C with a yield of 86 %. The sentences were corrected.
Q3.The DFT results should be listed the Supporting information. This makes it easy for the reader to repeat.
Reply: We have included our results of DFT computations in SI: cartesian coordinates of optimized structures together with electronic energies
Reviewer 2 Report
In this manuscript, authors reported single crystal structure of 2,3,4,5-tetraphenyl-1-monophosphaferrocene 2 and complex 3 from the product of reaction of W(CO)5 with complex 2 characterized by NMR, IR and elemental analysis. Authors also investigated electrochemical property of complexes 2 and 3, and ESR spectroscopy of complex 2. In addition, Authors also performed DFT calculation of complexes. I would like to recommend this study for publication in Molecules after minor revision.
1. In ESR, complex 3 did not show any signal, why?
2. In Fig. 2 and 4, please remove [7ua and added the ordinate and units.
3. Please elucidate why the CV is irreversible?
4. It is possible to obtain single crystal structure of complex 3 by changing crystalline method?
Author Response
Response to Reviewer 2 Comments
Manuscript ID: molecules-2222464
Title: Synthesis, structure, and electrochemical properties of 2,3,4,5-tetraphenyl-1-monophosphaferrocene derivatives
Reviewer #2 In this manuscript, authors reported single crystal structure of 2,3,4,5-tetraphenyl-1-monophosphaferrocene 2 and complex 3 from the product of reaction of W(CO)5 with complex 2 characterized by NMR, IR and elemental analysis. Authors also investigated electrochemical property of complexes 2 and 3, and ESR spectroscopy of complex 2. In addition, Authors also performed DFT calculation of complexes. I would like to recommend this study for publication in Molecules after minor revision.
Reply: First of all, we would like to thank the Reviewer for the detailed consideration of our manuscript, questions and remarks.
Q1. In ESR, complex 3 did not show any signal, why?
Reply: Such behavior of the complex 3 can be explained by various reasons. First, the relaxation time of the cation 3 can be shorter than that of the cation 2. Secondly, despite the fact that the calculations showed the preference of the low-spin state for both cationic forms of complexes, the probability of the high-spin state of the oxidized species 3 is slightly higher (ΔE cation 3 < ΔE cation 2). In the case of the high-spin state, we also do not observe ESR spectrum.
Other explanations can also be further chemical reactions of complex 3 such as dimerization.
We added the following sentence « Such behavior of complex 3 can be explained by the assumption that the relaxation time of the cation of 3 shorter than that of the cation of 2. »
Q2. In Fig. 2 and 4, please remove [7ua and added the ordinate and units.
Reply: Fig. 2 and 4 were corrected.
Q3. Please elucidate why the CV is irreversible?
Reply: In the manuscript, we described the possible causes of irreversible processes. In the case of compound 2, the oxidation process is quasi-reversible. In the case of complex 3, the CV is irreversible, since intramolecular rearrangement becomes impossible in the case of a charge change. The literature data describes that the solvent itself can also influence the irreversibility [Organometallics, Vol. 3, No. 8, 1984].
Q4. It is possible to obtain single crystal structure of complex 3 by changing crystalline method?
Reply: The authors acknowledge the reviewer's meaningful remark. We have tried different crystallization techniques to obtain single crystal structure of complex 3: hot recrystallization, cold crystallization and diffusion experiments and different solvents. Unfortunately, all attempts were unsuccessful. The structure of 3 was undoubtedly confirmed by multinuclear NMR spectroscopy and elemental analysis
Reviewer 3 Report
The authors describe the synthesis, spectroscopic characterization, electrochemical properties, and results of the theoretical calculations of a new sandwich mono-cyclopentadienyl iron(II) compound [FeCp(η5-PC4Ph4)] (2) with tetraphenyl-substituted monophosphacyclopentadienide ligand and its complex (3) with W(CO)5. This work is a development of the researches of this scientific group. Both compounds were properly characterized. The crystal structure of [FeCp(η5-PC4Ph4)] was determined by means of single-crystal X-ray analysis. The [FeCp(η5-PC4Ph4)] undergoes one oxidation and one reduction wave, while its complex [FeCp(η5-PC4Ph4)]W(CO)5 shows two oxidation and two reduction waves. I have found this manuscript to be interesting for a wide reading audience of the Journal. At the same time, some moments in the manuscript need to be explained more in details and some - to be corrected.
1. The electrochemical investigations.
- Table 2. The difference between Eox1 and 1/2Eox1 is not clear. The first one is a peak potential while the second one is a half-wave potential? In this case, may be it would be better to combine them in one column using the caption Eox1 (1/2Eox1), V for this column?
- Also, it is not clear wat these two potentials mean for complex 3 (0.85 and 0.35 V as given in Table 2). The figure 2 cannot answer this question because there is no potential switch above 0.8 V.
- Table 2. The 1/2Ered2 = 2.06 V for 3 should be -2.06 V
- lines 166-168. The authors say “The quasi-reversibility during the oxidation of structure 2 can be associated with the fact that during the formation of Fe(II) in Fe(III), the P-atom could be coordinated to the Fe-atom, as a result of which the re-reduction potential (0.16 V) is shifted to the negative region (Scheme 2).” However, there is no observable re-reduction peak at 0.16 V in Figure 2 . Please check.
- Please provide the current ratios Ia/Ic for al reversible/quasi-reversible processes in Table 2.
- Figure 2. What does 7uA means? Maybe, 7mA? The same question to Figure 3 (7ua).
- Scheme 2. The positive charge for two right structures is missed.
- the explanation of electrochemical behavior of 3 given on pages 7-8 (lines 214-218, Scheme 3) causes doubts. The reason is as follows: if the EC mechanism occurs at the first stage for 3, then the second oxidation wave (1/2Eox2) should be the same as for 2. But they are quite different (-2.06 and -2.25 V, respectively). Should be explained.
2. Please provide the full CV curves for both compounds in ESI.
3. Please provide the 1H, 13C and 31P NMR spectra for 2 and 3 in ESI.
4. A couple of minor moments:
- the reference list. Starting from the ref. 35, the volume numbers are given together with book number in this volume for a lot of references, e.g. ref. 35 “… Organometallics 2013, 32 (17), 4997–5000….” However, before ref. 35 these numbers are omitted. Please consult with the references rules of the Journal.
5. Line 56. “Existing synthetic methods allow obtaining various phospholide anions in pure form…” it is better to use “… allow to obtain …”
Author Response
Response to Reviewer 3 Comments
Manuscript ID: molecules-2222464
Title: Synthesis, structure, and electrochemical properties of 2,3,4,5-tetraphenyl-1-monophosphaferrocene derivatives
Reviewer #3 The authors describe the synthesis, spectroscopic characterization, electrochemical properties, and results of the theoretical calculations of a new sandwich mono-cyclopentadienyl iron(II) compound [FeCp(η5-PC4Ph4)] (2) with tetraphenyl-substituted monophosphacyclopentadienide ligand and its complex (3) with W(CO)5. This work is a development of the researches of this scientific group. Both compounds were properly characterized. The crystal structure of [FeCp(η5-PC4Ph4)] was determined by means of single-crystal X-ray analysis. The [FeCp(η5-PC4Ph4)] undergoes one oxidation and one reduction wave, while its complex [FeCp(η5-PC4Ph4)]W(CO)5 shows two oxidation and two reduction waves. I have found this manuscript to be interesting for a wide reading audience of the Journal. At the same time, some moments in the manuscript need to be explained more in details and some - to be corrected.
Reply: First of all, we would like to thank the Reviewer for the detailed consideration of our manuscript, questions and remarks.
Q1. The electrochemical investigations.
- Table 2. The difference between Eox1 and 1/2Eox1 is not clear. The first one is a peak potential while the second one is a half-wave potential? In this case, may be it would be better to combine them in one column using the caption Eox1 (1/2Eox1), V for this column?
Reply: Thank you for this comment. This typo has been corrected.
- Also, it is not clear wat these two potentials mean for complex 3 (0.85 and 0.35 V as given in Table 2). The figure 2 cannot answer this question because there is no potential switch above 0.8 V.
Reply: Thank you for this comment. This typo has been corrected. (0.85 is the potential relative to Ag/AgCl)
- Table 2. The 1/2Ered2 = 2.06 V for 3 should be -2.06 V
Reply: Thank you for this comment. It was corrected.
- lines 166-168. The authors say “The quasi-reversibility during the oxidation of structure 2 can be associated with the fact that during the formation of Fe(II) in Fe(III), the P-atom could be coordinated to the Fe-atom, as a result of which the re-reduction potential (0.16 V) is shifted to the negative region (Scheme 2).” However, there is no observable re-reduction peak at 0.16 V in Figure 2 . Please check.
Reply: As in remark 2, this is a typo. Initially, the data were vs. Ag/AgCl. All data was verified and corrected. Thank you for your comment.
- Please provide the current ratios Ia/Ic for al reversible/quasi-reversible processes in Table 2.
Reply: Current ratios Ia/Ic for al reversible/quasi-reversible processes were provided in Table 2.
- Figure 2. What does 7uA means? Maybe, 7mA? The same question to Figure 3 (7ua).
Reply: The value of the current is directly proportional to the concentration of the substance, and depends on the area of the electrode. The values given are correct. Taking into account the remark of another reviewer, the species of the CV were changed.
- Scheme 2. The positive charge for two right structures is missed.
Reply: The authors acknowledge the referee's meaningful remark. We have corrected the Scheme 2 with the conversion of Fe(II) to Fe(III).
- the explanation of electrochemical behavior of 3 given on pages 7-8 (lines 214-218, Scheme 3) causes doubts. The reason is as follows: if the EC mechanism occurs at the first stage for 3, then the second oxidation wave (1/2Eox2) should be the same as for 2. But they are quite different (-2.06 and -2.25 V, respectively). Should be explained.
Reply: Thank you for your comment. We changed the interpretation. «In the literature [Journal of Organometallic Chemistry, 295 (1985) 189-197], the reduction of the W(CO)5 complex of 3,3',4,4'-tetramethyl-1,1'-diphosphaferrocene was accompanied by an electrochemical-chemical mechanism. In our case, with only one phospholide ligand, this mechanism is not implemented, despite the fact that two reduc-tion waves are also observed, since in this case the second reduction wave doesn’t coin-cide with phosphaferrocene 2. The first reduction wave can be attributed to the formation of a radical anion on the phospholide anion (Scheme 3). The shift of the potential in com-parison with 2 to the anodic region is associated with the shift in the electron density from the phosphaferrocene fragment to the W(CO)5 fragment. The second reduction wave probably refers to the localization of the electron near on the W(CO)5 fragment and under the experimental conditions has time to be fixed without decomposition.»
Q2. Please provide the full CV curves for both compounds in ESI.
Reply: Thank you for your comment. We added additional CV data to ESI.
Q3. Please provide the 1H, 13C and 31P NMR spectra for 2 and 3 in ESI.
Reply: We added all NMR spectra for 2 and 3 in ESI.
Q4. A couple of minor moments:
- the reference list. Starting from the ref. 35, the volume numbers are given together with book number in this volume for a lot of references, e.g. ref. 35 “… Organometallics 2013, 32 (17), 4997–5000….” However, before ref. 35 these numbers are omitted. Please consult with the references rules of the Journal.
Reply: The references were corrected.
Q5. Line 56. “Existing synthetic methods allow obtaining various phospholide anions in pure form…” it is better to use “… allow to obtain …”
Reply: Thanks you. The sentence was corrected.